# Comparative Analysis of Myocardial Viability Multimodality Imaging in Patients with Previous Myocardial Infarction and Symptomatic Heart Failure

**DOI:** 10.3390/medicina58030368

**Published:** 2022-03-01

**Authors:** Egle Kazakauskaite, Donatas Vajauskas, Ruta Unikaite, Ieva Jonauskiene, Agneta Virbickiene, Diana Zaliaduonyte, Tomas Lapinskas, Renaldas Jurkevicius

**Affiliations:** 1Cardiology Clinic, Medical Academy, University of Health Sciences, 44307 Kaunas, Lithuania; ruta.unikaite@kaunoklinikos.lt (R.U.); ieva.jonauskiene@kaunoklinikos.lt (I.J.); agneta.virbickiene@kaunoklinikos.lt (A.V.); diana.zaliaduonyte@kaunoligonine.lt (D.Z.); tomas.lapinskas@kaunoklinikos.lt (T.L.); renaldas.jurkevicius@kaunoklinikos.lt (R.J.); 2Kaunas Region Society of Cardiology, 44307 Kaunas, Lithuania; 3Radiology Clinic, Medical Academy, University of Health Sciences, 44307 Kaunas, Lithuania; donatas.vajauskas@kaunoklinikos.lt

**Keywords:** myocardial viability, SENC imaging, 18F-fluorodeoxyglucose positron emission tomography, late gadolinium enhancement, reversibility score

## Abstract

*Background and Objectives*: To compare the accuracy of multimodality imaging (myocardial perfusion imaging with single-photon emission computed tomography (SPECT MPI), 18F-fluorodeoxyglucose positron emission tomography (18F-FDG PET), and cardiovascular magnetic resonance (CMR) in the evaluation of left ventricle (LV) myocardial viability for the patients with the myocardial infarction (MI) and symptomatic heart failure (HF). *Materials and Methods*: 31 consecutive patients were included in the study prospectively, with a history of previous myocardial infarction, symptomatic HF (NYHA) functional class II or above, reduced ejection fraction (EF) ≤ 40%. All patients had confirmed atherosclerotic coronary artery disease (CAD), but conflicting opinions regarding the need for percutaneous intervention due to the suspected myocardial scar tissue. All patients underwent transthoracic echocardiography (TTE), SPECT MPI, 18F-FDG PET, and CMR with late gadolinium enhancement (LGE) examinations. Quantification of myocardial viability was assessed in a 17-segment model. All segments that were described as non-viable (score 4) by CMR LGE and PET were compared. The difference of score between CMR and PET we named reversibility score. According to this reversibility score, patients were divided into two groups: Group 1, reversibility score > 10 (viable myocardium with a chance of functional recovery after revascularization); Group 2, reversibility score ≤ 10 (less viable myocardium when revascularisation remains questionable). *Results*: 527 segments were compared in total. A significant difference in scores 1, 2, 3 group, and score 4 group was revealed between different modalities. CMR identified “non-viable” myocardium in 28.1% of segments across all groups, significantly different than SPECT in 11.8% PET in 6.5% Group 1 (viable myocardium group) patients had significantly higher physical tolerance (6 MWT (m) 3892 ± 94.5 vs. 301.4 ± 48.2), less dilated LV (LVEDD (mm) (TTE) 53.2 ± 7.9 vs. 63.4 ± 8.9; MM (g) (TTE) 239.5 ± 85.9 vs. 276.3 ± 62.7; LVEDD (mm) (CMR) 61.7 ± 8.1 vs. 69.0 ± 6.1; LVEDDi (mm/m^2^) (CMR) 29.8 ± 3.7 vs. 35.2 ± 3.1), significantly better parameters of the right heart (RV diameter (mm) (TTE) 33.4 ± 6.9 vs. 38.5 ± 5.0; TAPSE (mm) (TTE) 18.7 ± 2.0 vs. 15.2 ± 2.0), better LV SENC function (LV GLS (CMR) −14.3 ± 2.1 vs. 11.4 ± 2.9; LV GCS (CMR) −17.2 ± 4.6 vs. 12.7 ± 2.6), smaller size of involved myocardium (infarct size (%) (CMR) 24.5 ± 9.6 vs. 34.8 ± 11.1). Good correlations were found with several variables (LVEDD (CMR), LV EF (CMR), LV GCS (CMR)) with a coefficient of determination (R^2^) of 0.72. According to the cut-off values (LVEDV (CMR) > 330 mL, infarct size (CMR) > 26%, and LV GCS (CMR) < −15.8), we performed prediction of non-viable myocardium (reversibility score < 10) with the overall percentage of 80.6 (Nagelkerke R^2^ 0.57). *Conclusions*: LGE CMR reveals a significantly higher number of scars, and the FDG PET appears to be more optimistic in the functional recovery prediction. Moreover, using exact imaging parameters (LVEDV (CMR) > 330 mL, infarct size (CMR) > 26% and LV GCS (CMR) < −15.8) may increase sensitivity and specificity of LGE CMR for evaluation of non-viable myocardium and lead to a better clinical solution (revascularization vs. medical treatment) even when viability is low in LGE CMR, and FDG PET is not performed.

## 1. Introduction

There are two ways that dysfunctional but viable myocardium can exist. Myocardial stunning was firstly described by Braunwald and Kloner as “prolonged, post-ischemic ventricular dysfunction that occurs after brief periods of nonlethal ischemia” [1]. Meanwhile, hibernating myocardium [2,3,4] was first described as the result of repetitive and long-lasting ischemia due to significant coronary artery stenosis and severely limited coronary flow reserve. If the myocardium is in the hibernation stage, the contractility may recover spontaneously when the blood flow is re-established [5].

Several imaging techniques have been proposed to assess viable myocardium: dobutamine stress or myocardial contrast echocardiography, myocardial perfusion imaging with single-photon emission computed tomography (SPECT MPI), 18F-fluorodeoxyglucose (18F-FDG) positron emission tomography (PET), cardiovascular magnetic resonance (CMR) with late gadolinium enhancement (LGE), and computed tomography (CT). These techniques describe different characteristics of dysfunctional but viable myocardium by evaluating myocardial perfusion, cell membrane integrity, mitochondria, glucose metabolism, scar tissue, and contractile reserve.

SPECT MPI with 99mTc-labeled sestamibi demonstrates the myocardial uptake and retention of sestamibi, which are dependent on perfusion, cell membrane integrity, and mitochondrial function. SPECT MPI is the frequently used imaging technique for the detection of coronary artery disease (CAD) and demonstrates excellent sensitivity in the diagnosis of hemodynamically significant coronary artery stenosis; however, its specificity remains questionable due to certain limitations, such as motion artifact [6], excessive subdiaphragmatic activity [7], breast attenuation [8], and asymmetric ventricular wall thickening [9].

Cardiac PET is a precise non-invasive imaging modality that estimates the cellular function of the heart, while other non-invasive cardiovascular imaging modalities are based on the evaluation of heart anatomy, morphology, structure, function, and tissue characteristics [10]. In clinical cardiology, PET is established as the best available test for the assessment of myocardial viability, as the standard for validation of many new techniques [11,12]. For the evaluation of myocardial viability, various tracers have been used in combination with PET -11C-acetate and 82Rb; however, the most frequently used tracer is 18F-FDG. Comparing 18F-FDG images with the myocardial perfusion images enables distinguishing viable myocardium from a fibrotic scar.

CMR is a highly efficient method for the assessment of myocardial function and myocardial scar [13,14,15]. Due to its high spatial resolution, CMR stands out for its superior imaging quality and capacity to reveal ischemic areas that would not be detectable by other imaging modalities. Several different CMR methods may be applied for the evaluation of left ventricle (LV) function, myocardial scar, and viable myocardium.

LGE CMR [15] imaging allows differentiation between transmural and subendocardial necrosis with high sensitivity and provides a negative predictive value (NPV) for predicting improved segmental LV contractile function after revascularization [15,16,17,18].

Nevertheless, the strain-encoded CMR (SENC) is proposed as the novel CMR method of myocardial deformation that reveals the objective description of myocardial function and indicates the evaluation of regional myocardial deformation expressed as regional myocardial strain [19,20,21]. Significant advantages of SENC quantitative analysis (substantially less time-consuming [20], high-temporal resolution, and no need for contrast application) may influence handling this novel method in the clinical routine.

The variety of different imaging modalities for detecting viable myocardium leads not only to a better understanding of hibernating myocardium and clinical decision making but also may be confusing trying to find the best imaging modality for the exact patient. The data of “head-to-head” comparison is lacking because all modalities are associated with different limitations: time consumption (both clinician and patient), radiation exposure (nuclear medicine imaging), excessive cost, contraindications. The idea of this study was not only to compare different non-invasive imaging modalities for patients with significantly reduced LV systolic function and previous myocardial infarction (MI) but also to find imaging parameters that may predict a sufficient quantity of viable myocardium for the recovery of contractility in terms of successful myocardial revascularization, that could be potentially assessed for the exact patient in the clinical routine. 

## 2. Methods

### 2.1. Study Population

We prospectively recruited 31 consecutive patients referred to the Cardiology department of Lithuania Health Sciences University Hospital Kaunas Clinics. The Inclusion criteria were a medical history of previous MI, symptomatic heart failure (HF) with New York Heart Association (NYHA) functional class II, III, or IV, reduced LV ejection fraction (EF), ≤40 percentage (%), and expanded coronary artery atherosclerosis evaluated by invasive coronary angiography when the need of percutaneous intervention was questionable because of suspected myocardial scar tissue.

Clinical data from medical documentation: age, gender, comorbidities, medications, symptoms of heart failure or other diseases, risk factors of cardiovascular disease, data of objective investigation (weight, height, calculated body surface area, and body mass index, heart rate, arterial blood pressure), and electrocardiographic findings were collected and analyzed. All patients underwent physical examination, six-minute walk test (6 MWT) evaluation, conventional 2D echocardiography at rest, rest SPECT MPI/ FDG PET imaging, and CMR with LGE and SENC. The CMR and nuclear cardiology imaging analysis were performed by one experienced physician dedicated to the exact imaging modality in our institution.

### 2.2. Transthoracic Echocardiography

Transthoracic echocardiography (TTE) was performed in all patients. All measurements were obtained according to the valid guidelines (the European Society of Cardiology (ESC), the European Association for Cardio-Thoracic Surgery (EACTS), and American Heart Association (AHA)) [22]. The conventional transthoracic 2D echocardiography system (EPIQ 7, Phillips Ultrasound, Inc., Bothell, WA, USA) with 1.5–4.6 MHz transducer was used. We evaluated all morphometric assessments of the chambers using 2D-guided linear and tracing (for volumes evaluation) measurements. In the parasternal long-axis view, we performed linear measurements of the left heart: the anteroposterior diameter of the left atrium (LA) and LV end-diastolic diameter (LVEDD), LV end-systolic diameter (LVESD). Evaluation of the volumes (LV end-diastolic volume (LVEDV), LV end-systolic volume (LVESV)) was achieved by biplane apical tracing. LV EF was calculated using modified Simpson’s biplane method (in the apical four- and two-chamber views, LV endocardial borders at end-diastole and end-systole were manually traced). The function of RV was evaluated by measuring the tricuspid annular plane systolic excursion (TAPSE) and the peak systolic velocity of the tricuspid annulus (RV S’). TAPSE was obtained using the M-mode in the apical four-chamber view, while RV S’ was obtained by tissue doppler imaging.

### 2.3. Cardiovascular Magnetic Resonance

#### 2.3.1. CMR: Study Protocol

All CMR images were acquired using a 1.5 Tesla scanner (Siemens Magnetom Aera, Siemens AG Healthcare Sector, Erlangen, Germany) with an 18-channel phased array coil in a supine position. The study protocol included an initial survey to define imaging planes.

Cine images were acquired using retrospectively gated balanced steady-state free precession (SSFP) sequence with short periods of breath-holding in three LV long-axis (2-chamber, 3-chamber, and 4-chamber) planes. The ventricular 2-chamber and 4-chamber planes were used to plan a contiguous stack of short-axis slices covering the entire LV. The in-plane resolution of cine images was 0.9 × 0.9 mm, slice thickness of 8 mm with 2 mm interslice gap, and 25 phases per cardiac cycle. The following parameters were used: repetition time (TR) = 3.3 ms, echo time (TE) = 1.6 ms, flip angle = 60°, acquisition voxel size = 1.8 × 1.8 × 8.0 mm^3^, and 25 phases per cardiac cycle.

The late gadolinium enhancement (LGE) images were obtained 10 min after the injection of 0.15 mmol/kg gadobutrol (Gadovist^®^, Bayer Schering Pharma AG, Berlin, Germany). A Look-Locker sequence was acquired to determine the inversion time to null the signal of the LV myocardium. A 2D inversion recovery fat-saturated spoiled gradient-echo sequence was used to detect scar tissue in three LV long-axis and short-axis orientations.

#### 2.3.2. CMR: Image Analysis

All cine images were analyzed offline using Medis Suite, version 3.0 (Leiden, The Netherlands) under a recent consensus document for quantification of LV function and mass using CMR [23]. The end-diastolic and end-systolic cardiac phases were detected visually. After manual contouring of endocardial and epicardial borders, LV end-diastolic (LVEDV), LV end-systolic (LVESV), LV ejection fraction (LVEF), and LV end-diastolic mass (LVEDM) were calculated. Papillary muscles were considered part of the blood pool. LV volumes and myocardial mass were adjusted to body surface area, determined using the Mosteller equation.

The endocardial and epicardial contours drawn on cine images were transferred into LGE images. The presence and extent of LGE were quantified using the signal threshold versus reference mean (STRM) > 3 standard deviations (SD) method as it provides the highest accuracy with acceptable reproducibility compared with other signal intensity threshold techniques [24]. The total LGE volume and mass were calculated automatically.

For quantification of myocardial viability, all short-axis images were segmented using a 17-segment model. The LGE extent was assessed and quantified in each short-axis segment by the 5-point scoring system: score 0 = LGE, viable myocardium; score 1 = LGE 0–25% of wall thickness; score 2 = LGE 25–50% of wall thickness; score 3 = LGE 50–75% of wall thickness; score 4 = transmural scar (no viability).

### 2.4. Cardiac Nuclear Medicine Imaging 

SPECT MPI and FDG PET viability analysis were performed by a single-blinded investigator. Myocardial perfusion and viability images were reconstructed and analyzed in the 17-segment model.

#### 2.4.1. SPECT MPI: Study Protocol

A rest-only ECG gated SPECT myocardial perfusion imaging was performed in our study. All patients were intravenously injected with a technetium 99m (99mTc) labeled sestamibi, adjusted to body mass (280 ± 65 MBq). ECG gated SPECT MPI imaging was performed 60 min after 99mTc-MIBI injection, with a dual-head BrightView XCT (Philips healthcare) gamma camera, using a low-energy, high-resolution collimator, a 20% window at 140 keV, 64 × 64 matrix, an orbit with 120 projections, at 3-degree steps and a 20 s per step. SPECT MPI acquisition was performed in patients positioned in a supine position with the arms held above the head.

#### 2.4.2. SPECT MPI: Image Analysis

Gated and non-gated SPECT MPI images were reconstructed using OSEM iterative reconstruction with the dedicated Philips workstation. Analysis was performed using the Cedars-Sinai QGS/QPS software package. Images of the left ventricle were analyzed in short, vertical, and horizontal long-axis and polar maps.

Perfusion defects in each LV 17 segments were scored using a 5-point scoring system as follows: 0—normal perfusion, 1—equivocal or mild reduction in perfusion, 2—moderately reduced perfusion, 3—severely reduced perfusion, and 4—absent perfusion/perfusion defect.

The summed rest score (SRS) was obtained in all LV by summing the scores of the corresponding 17 segments.

#### 2.4.3. FDG PET: Study Protocol

ECG gated FDG PET myocardial viability imaging was performed in our study. All patients were intravenously injected with an 18F-FDG adjusted to the body mass (4 MBq/kg) after fasting for at least 6 h. The following oral glucose and insulin loading protocols were used in all patients. First, the patient’s glucose level was measured. In nondiabetic patients, 50 g oral glucose was administered if blood glucose level was 8.33 mmol/L and below, and 25 g glucose if blood glucose level was 8.33–13.8 mmol/L. The physician was notified if blood glucose was > 11.11 mmol/L. In diabetic patients, 25 g glucose was administered if blood glucose was 8.33 mmol/L and below, and 12.5 g glucose for blood glucose of 8.33–13.8 mmol/L. No glucose was administered if blood glucose was > 13.8 mmol/L. Thirty minutes after glucose load, blood glucose level was measured, if blood glucose level was below 7.77 mmol/L, repetitive blood glucose level was measured after 15 min, and 18F-FDG was injected intravenously. If the blood glucose level was 7.78 mmol/L or more, adjusted to the blood glucose level, an insulin injection was made, and the blood glucose level measurement was repeated.

PET images were obtained 60 min after 18F-FDG injection using Discovery XCT PET/CT (Ge Medical Systems, Chicago, IL, USA). The duration of PET acquisition was 15 min following a low-dose CT scan for attenuation correction. 

### 2.5. PET, MPI, and CMR Comparative Analysis

SPECT MPI and FDG PET viability images were compared visually using short axis, horizontal, and vertical long-axis images as well as polar maps. 

On FDG PET images, distribution of 18F-FDG in areas of SPECT MPI perfusion defect segments was assessed. Myocardial viability was assessed with visual analysis scoring according to SPECT MPI defect reversibility on FDG PET viability imaging. Score 4 was evaluated as non-viable myocardium, score 1–3 was described as viable myocardium with a chance of functional recovery after revascularization, and score 0 was reported as viable myocardium with correspondence to SPECT MPI score 0 (normal perfusion). 

Based on the SPECT and PET findings, LV segments were divided into three groups. Group 1 included SPECT MPI normal/PET viable myocardium (score 0), Group 2 included SPECT MPI minimal to severe perfusion defect/FDG PET viable myocardium (score 1–3), and Group 3 included SPECT MPI absent perfusion/perfusion defect/FDG PET non-viable myocardium (score 4).

Understanding the different backgrounds of myocardial viability evaluation with imaging modalities (CMR LGE describes scar tissue, FDG PET shows metabolism of the viable myocytes), we compared all segments that were described as non-viable (score 4) by CMR LGE and FDG PET. The difference of score between CMR and PET we named reversibility score. According to this reversibility score, we divided patients into two groups: Group 1, reversibility score > 10 (viable myocardium with a chance of functional recovery after revascularization); Group 2, reversibility score ≤ 10 (less viable myocardium when revascularization remains questionable).

### 2.6. Statistical Analysis

Quantitative variables were expressed as mean (M) ± standard deviation (SD) and categorical variables as frequencies and percentages. The Student’s *t*-test was used for comparisons of continuous variables. The quantitative variable’s normality assumption was verified using the Kolmogorov–Smirnov test. Following the probability distribution of the variables, Pearson’s or Spearmen’s correlation coefficient was used.

We focused the analytic strategy on predicting viable or non-viable (according to Reversibility Score) myocardium from a set of readily available patient characteristics. Comparisons of patient characteristics between groups were based on the chi-square test (for categorical variables) and t-test or Mann–Whitney-U test (quantitative variables).

Univariate and multivariate logistic regression analyses were performed to determine risk factors for no viable myocardium. Clinically relevant variables with *p* < 0.05 on univariate analysis were incorporated into the multivariate models. Stepwise logistic regression was performed to identify independent predictors of non-viable myocardium. Results were reported as effect sizes [odds ratios (ORs)] with 95% confidence intervals (CIs). Logistic regression was used to reduce the set of patient characteristics into a probability of developing the outcome of interest, non-viable myocardium. These model-based probabilities were analyzed using receiver operating characteristic (ROC) curves. ROC curves are a graphic representation of the trade-off between the false-negative and false-positive rates for every possible cut-off probability of non-viable myocardium. The area under the curve (AUC) summarizes information about the outcome contained in the ‘‘predictor’’ set, with a value of 1 as maximum. All reported *p-*values were two-sided, and *p*-values of < 0.05 were considered to indicate statistical significance. SPSS software (IBM, SPSS Statistics, Version 22) was used for statistical testing.

## 3. Results

### 3.1. Study Population

Demographic and clinical characteristics of patients involved in the study are listed comprehensively in Table 1. The mean age of the patients was 61.6 ± 10.3 and the majority of them were men (n = 28, 90.3%). Among the different risk factors, hypertension (n = 30, 96.8%) and hyperlipidemia (n = 31, 100%) were the most often observed in the study population. The dominant NYHA functional class was 2nd (n = 15, 48.4%) and 3rd (n = 15, 48.4%) despite the optimal medical treatment. Most of the patients were suffering from dyspnea (n = 29, 93.5%). All patients had hemodynamically significant CAD; moreover, analysis of coronary artery lesions revealed that left anterior descending (LAD) artery disease was found in the majority of the study population (n = 21, 67.7%). Furthermore, the comorbidities were evaluated: previous stroke (n = 2, 6.5%), chronic obstructive pulmonary disease (n = 1, 3.2%), asthma (n = 1, 3.2%), previous pulmonary thromboembolism (n = 3, 9.7%), atrial fibrillation or flutter (n = 6, 19.4%), pacemaker (n = 3, 9.7%), oncological disease (n = 3, 9.7%), and significant valvular heart disease (n = 14, 45.2%).

### 3.2. Cardiovascular Imaging 

As mentioned previously, all patients of the study underwent CMR and nuclear medicine imaging: MPI SPECT and FDG PET. 

For quantification of myocardial viability, all short-axis LV images of different imaging modalities (SPECT, PET, and CMR) were segmented using a 5-segment model.

A total of 527 segments were compared. A 0 score showed normal perfusion in SPECT (n = 299), viable myocardium in PET (n = 333), and no myocardial scar in LGE CMR (n = 279). Scores 1, 2, 3 were determined as decreased myocardial perfusion but still viable myocardium with a chance of functional recovery after revascularization (SPECT n = 166, PET n = 160, CMR n = 100). A 4 score was evaluated as “non-viable” myocardium without beneficial functional prognosis (no perfusion in SPECT (n = 62), no myocardial FDG-uptake in PET (n = 34), and transmural scar in CMR images (n = 148)). We found a significant difference in scores 1, 2, 3, and 4 groups between different modalities (Table 2, Figure 1). In terms of functional improvement estimation after revascularization, CMR identified “non-viable” myocardium in 28.1% of segments across all groups, SPECT in 11.8% (*p* < 0.05), PET in 6.5% (*p* < 0.05). 

As we noticed significant difference between imaging modalities in evaluating non-viable myocardium, we performed only 4 score (transmural scar) analyses in both modalities: PET and CMR LGE (Figure 2). The same segments of score 4 were detected by PET and CMR in the majority of cases (n = 31, 91.2%), while few segments from 4 score PET changed to 3 score CMR LGE (n = 3, 8.8%). In the CMR LGE group, score 4 (n = 148, 100%) distribution compared to PET was different: n = 39 (26.4%) turned to PET 0 score, n = 27 (18.2%) to 1 score, n = 26 (17.6%) to score 2, n = 25 (16.9%) to score 3, n = 31 (20.9%) to score 4. We summed the difference of scores between the modalities per patient as it is mentioned previously and divided patients in two groups: group 1—reversibility score > 10, viable myocardium; group 2—reversibility score ≤ 10, insufficient amount of viable myocardium. Comparison of these two groups revealed that in Group 1 (viable myocardium group) patients had higher physical tolerance (6 MWT (m) 389.2 ± 94.5 vs. 301.4 ± 48.2; *p* < 0.05), less dilated LV (LVEDD (mm) (TTE) 53.2 ± 7.9 vs. 63.4 ± 8.9, *p* < 0.05; MM (g) (TTE) 239.5 ± 85.9 vs. 276.3 ± 62.7, *p* < 0.05; LVEDD (mm) (CMR) 61.7 ± 8.1 vs. 69.0 ± 6.1, *p* < 0.05; LVEDDi (mm/m^2^) (CMR) 29.8 ± 3.7 vs. 35.2 ± 3.1, *p* < 0.05), better parameters of the right heart (RV diameter (mm) (TTE) 33.4 ± 6.9 vs. 38.5 ± 5.0, *p* < 0.05; TAPSE (mm) (TTE) 18.7 ± 2.0 vs. 15.2 ± 2.0, *p* < 0.05), better LV SENC function (LV GLS (CMR) −14.3 ± 2.1 vs. 11.4 ± 2.9, *p* < 0.05; LV GCS (CMR) −17.2 ± 4.6 vs. −12.7 ± 2.6, *p* < 0.05), smaller size of involved myocardium (infarct size (%) (CMR) 24.5 ± 9.6 vs. 34.8 ± 11.1, *p* < 0.05) (Table 3). The reversibility score (counted from the difference of CMR LGE 4 score to any PET score) well correlated with all imaging parameters (Table 4). Using a linear regression model, we revealed the correlation between independent factor, reversibility score, and several variables (LVEDD (CMR), LV EF (CMR), LV GCS (CMR)) with the coefficient of determination (R^2^) 0.72 (Table 5). Moreover, we performed ROC analyses to determine the cut-off values of selected clinical and imaging variables for viable or non-viable myocardium. The findings of this analysis are given in Table 6. According to the cut-of values the multiple logistic regression model for prediction of non-viable myocardium (reversibility score < 10) was performed (Table 7). The LVEDV (CMR) > 330 mL, infarct size (CMR) > 26%, and LV GCS (CMR) < −15.8 predicted the reversibility score < 10 with an overall percentage of 80.6 (Nagelkerke R^2^ 0.57). 

## 4. Discussion

The most important finding of our study is the Reversibility Score (CMR and PET-based score model), which demonstrated better clinical and imaging characteristics for patients with a > 10 score difference between PET and CMR viability evaluation (reversibility score was calculated from the difference of CMR LGE 4 score to any PET score). Moreover, according to the cut-of values of selected clinical and imaging variables for viable or non-viable myocardium our study demonstrated prediction of non-viable myocardium using exact imaging parameters (LVEDV (CMR) > 330 mL, infarct size (CMR) > 26% and LV GCS (CMR) < −15.8). It may lead to the conclusion that despite transmural scar on LGE CMR, myocardium still may be viable if the mentioned CMR parameters are described as noticed previously. This added value of exact imaging parameters can change the clinical strategy of the patient and avoid HF progression by precisely selecting patients for revascularization even when viability is low in LGE CMR and PET is not performed.

Moreover, to our knowledge, the present study is the first to perform a “head-to-head” comparison of not only SPECT, PET, and CMR imaging for the evaluation of myocardial viability, but also application of CMR SENCE as the additional tool to increase sensitivity and specificity of LGE CMR. In our study LGE CMR showed a significantly higher number of scars associated with a low likelihood for functional improvement after revascularization, which might prevent patients from unnecessary invasive procedures and potential risks.

Hunold et al. [25] performed the “head-to-head” analysis of myocardial viability assessment with PET and LGE CMR with different LV systolic function and concluded that if the LV function is severely or moderately reduced, CMR detects considerably more myocardial scars than PET and is generally less optimistic concerning functional recovery after revascularization.

The explanation for this disagreement might be the higher spatial resolution of CMR compared to PET [26,27]. Moreover, studies are demonstrating that LGE CMR can detect a minimum of 2 g of irreversibly injured myocardium while PET requires at least 10 g of myocardial tissue [28]. Another disadvantage of nuclear imaging is radiation exposure, long examination time, and the necessity of appropriate tracers which have relatively short physical half-time. However, FDG PET is not associated with LGE CMR contraindications such as implantable cardiac devices, liver, or renal insufficiency.

Literature shows that the value of viability testing in the clinical routine remains controversial [29]. Several studies consider myocardial viability as an independent predictor of cardiac death in ischemic cardiomyopathy patients who underwent conservative medical treatment alone [30]. Allman et al. [31] demonstrated the strong association between myocardial viability and improved survival after revascularization therapy. Other studies suggested that a significant extent of viable myocardium (more than 10%) is associated with increased survival if the patient is undergoing revascularisation therapy compared to medical treatment alone [32]. However, the data is conflicting. Three prospective randomized trials with patients suffering from chronic ischemic heart disease (IHD) and HF with decreased LV EF were performed: PET and recovery following revascularization (PARR-2) trial [33], the HF revascularization (HEART) trial [34], and the surgical treatment for ischemic HF (STICH) trial [35]. These trials failed to show a significant survival benefit of revascularization over optimal medical treatment alone in patients with confirmed viable myocardium. Nonetheless, the results of previously mentioned studies are still debated widely [36,37] with the agreement that limitations of the studies are significant and may negatively affect the results [37]. Therefore, despite the lack of significant correlation between myocardial viability and benefit from revascularization, recent ESC guidelines of the myocardial revascularisation [38] suggested that non-invasive imaging may be considered for the assessment of myocardial ischemia and viability in patients with HF and CAD before the decision on revascularization [31,32,35].

Literature shows conflicting data not only about the impact of myocardial viability on successful revascularisation but also about different non-invasive imaging methods which could be superior in detecting viable myocardium.

Diagnostic accuracy for detection of infarct transmurality by LGE CMR is a well-established [39]. However, it is also confirmed that for segments with the extent of non-viable myocardium of between 25% and 75% the grey zone exists and leads to a variable range for functional recovery after revascularization of between 10% and 64% [16,40,41]. Moreover, the extent of the “non-viable” myocardium may be overestimated because of the LGE presentation in salvaged myocardium [42]. Recent CMR studies are focused on the value of SENC for the assessment of myocardial viability which is less time-consuming and might be performed without a contrast agent [20,43,44,45,46]. Studies are demonstrating that SENC is increasing LGE CMR sensitivity and specificity (100% and 86% respectively) [43] and shows the ability to differentiate between subendocardial and transmural MI [45]. However, the SENC sequence is unfortunately not yet widely feasible in busy CMR centers (quantification analysis is requiring additional time and dedicated expert) despite the relatively large amount of scientific evidence of its incremental value for distinguishing viable and non-viable myocardium.

Therefore, our study shows not only the incremental value of reversibility score but also gives additional information that combining the results of different imaging modalities might lead to a better clinical solution. The results of our study suggest the clinical approach for myocardial viability evaluation, diminishing radiation exposure to the patient whenever it is possible firstly performing TTE and CMR. If the CMR results demonstrate a sufficient quantity of viable myocardium revascularisation should be performed. However, according to the results of our study, LGE CMR reveals a significantly higher number of scars, therefore FDG PET should be performed in cases when viability is doubtful in CMR, as FDG PET appears to be more optimistic in functional recovery prediction. This algorithm of myocardial viability evaluation could be applied in clinical practice, leading not only to the precise evaluation of myocardial viability guiding to the better clinical decision (medical treatment, revascularisation or heart transplantation), but also avoiding unnecessary radiation exposure to the patient.

The main limitation of our study was the small study sample. However, according to the study protocol, we have applied different imaging modalities for patients with ischaemic symptomatic HF and expanded coronary artery atherosclerosis when the need for revascularisation was doubtful due to suspected myocardial scar tissue. In this patient group, we were able to perform a head-to-head comparison of three imaging modalities (TTE, nuclear cardiology imaging and CMR) to evaluate viable myocardium. This approach was associated with time consumption, cost-effectiveness, radiation exposure and limited study population. Another important limitation of our study was a possible anatomical misalignment between different imaging modalities because the evaluation of the 17 segments model of the LV is slightly different compared to nuclear imaging and CMR. Solving this anatomical misalignment problem, the best solution could be a PET/CMR hybrid scan protocol offering a wide evaluation of the heart (anatomical by CMR and functional by PET), however, in our center, it is not technically applicable. Another limitation of our study is the absence of the follow-up, which could confirm our prognostic model of parameters for evaluation of the viable/non-viable myocardium and the benefit of possible revascularisation. However, the study population was too small, and clinicians were not always following the recommendations for revascularization vs. medical treatment. Therefore, the precise follow-up in our study was not possible.

## 5. Conclusions

LGE CMR reveals a significantly higher number of scars and the FDG PET appears to be more optimistic in functional recovery prediction. Moreover, using exact imaging parameters (LVEDV (CMR) > 330 mL, infarct size (CMR) > 26% and LV GCS (CMR) < −15.8) may increase sensitivity and specificity of LGE CMR for evaluation of non-viable myocardium and lead to a better clinical solution (revascularization vs. medical treatment) even when viability is low in LGE CMR and FDG PET is not performed.

## Figures and Tables

**Figure 1 medicina-58-00368-f001:**
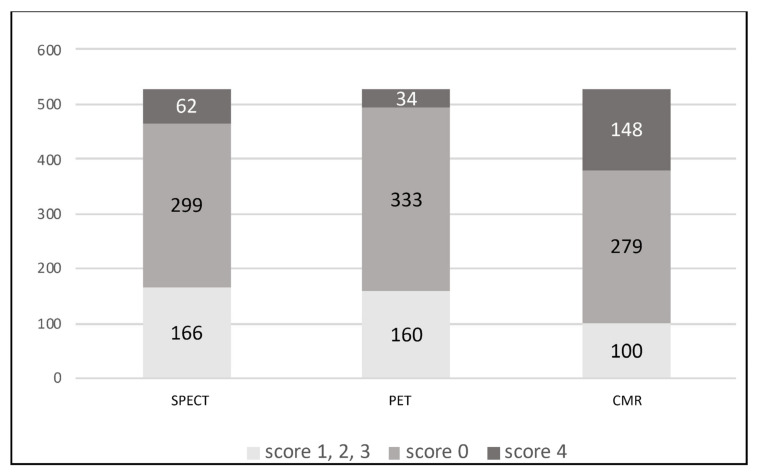
Segmental analysis of myocardial viability in 527 segments within different imaging modalities.

**Figure 2 medicina-58-00368-f002:**
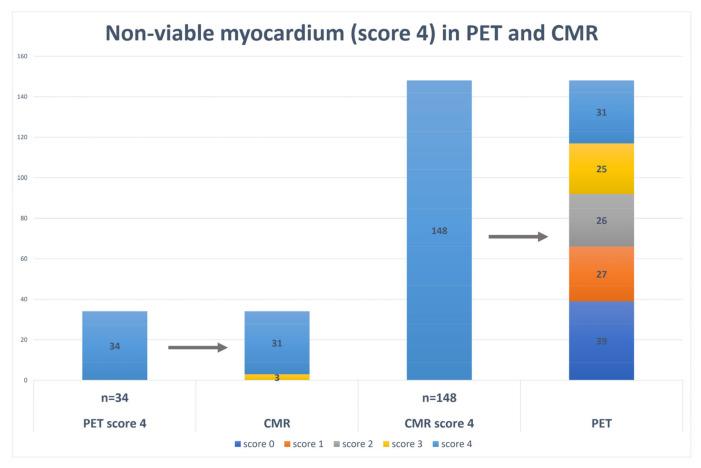
Non-viable myocardium (score 4) comparison within two imaging modalities (PET and CMR LGE).

**Table 1 medicina-58-00368-t001:** Baseline characteristics.

Baseline Characteristics	Total n = 31
Age in years, M ± SD	61.6 ± 10.3
Gender (Male/Female), n (%)	28 (9.3)/3 (9.7)
Coronary risk factors, n (%)	
- Hypertension,	30 (96.8)
- Diabetes,	2 (6.5)
- Obesity (BMI ≥ 30),	14 (45.2)
- Hyperlipidemia	31 (100.0)
- Smoking	18 (58.1)
- Family history of premature CAD	14 (45.2)
CAD history, n (%)	
- Prior STEMI/NSTEMI	31 (100.0)
- Prior PCI	21 (67.7)
- Prior CABG	2 (6.5)
- Conservative treatment	8 (25.8)
Comorbidities, n (%)	
- Stroke	2 (6.5)
- COPD	1 (3.2)
- Asthma	1 (3.2)
- Pulmonary thromboembolism	3 (9.7)
- Atrial fibrillation/flutter	6 (19.4)
- Pacemaker	3 (9.7)
- Oncological disease	3 (9.7)
Symptoms, n (%)	
- Angina	11 (35.5)
- Dyspnoea	29 (93.5)
- Others	6 (19.4)
NYHA class (M ± SD)	
- II, n (%)	15 (48.4)
- III, n (%)	15 (48.4)
- IV, n (%)	1 (3.2)
6 MWT, M ± SD	348.6 (81.9)
Coronary artery disease, culprit lesion, n (%)	
- LAD	21 (67.7)
- CX	1 (3.2)
- RCA	3 (9.7)
- LAD + CX	1 (3.2)
- LAD + RCA	3 (9.7)
- CX + RCA	2 (6.5)
Significant valvular heart disease, n (%)	14 (45.2)
- Aortic stenosis	0 (0)
- Aortic regurgitation	1 (3.2)
- Mitral stenosis	0 (0)
- Mitral regurgitation	13 (41.9)
- Tricuspidal regurgitation	4 (12.9)
Medications, n (%)	
- Aspirin	25 (80.6)
- Clopidogrel/Ticagrelor	11 (35.5)
- Anticoagulant	6 (19.4)
- ACEI/ARB	31 (100)
- Beta-blocker	30 (96.8)
- CCB	1 (3.2)
- Statin	31 (100.0)
- Nitrate	8 (25.8)
- Digoxin	3 (9.7)
- Ivabradine	9 (29.0)
- Spironolactone	24 (77.4)

Continuous variables expressed as mean (M) ± standard deviation (SD) for symmetric data. Categorical variables expressed as count and percentage of patients. Abbreviations: BMI, body mass index; CAD, coronary artery disease; STEMI, ST elevation myocardial infarction; NSTEMI, non-ST elevation myocardial infarction; PCI, percutaneous coronary intervention; CABG, coronary artery bypass grafting; COPD, chronic obstructive pulmonary disease; NYHA, New York Heart Association; 6 MWT, six-minute walk test; LAD, left anterior descending artery; CX, circumflex artery; RCA, right coronary artery; ACEI/ARB, angiotensin-converting enzyme inhibitors/angiotensin receptor blockers; CCB, calcium channel blockers.

**Table 2 medicina-58-00368-t002:** Comparative analysis of normal myocardial segments, subendocardial scar segments, and transmural scar segments between different imaging modalities (n = 527).

	Viable Myocardium (Score 0)			Subendocardial Scar (Score 1–3)			Transmural Scar (Score 4)		
			CMR			CMR			CMR
			279 (52.9%)			100 (19%)			148 (28.1%)
SPECT		299 (56.7%)			166 (31.5%)			62 (11.8%)	
*p*-value		*p* = 0.11			*p* < 0.05			*p* < 0.05	
PET	333 (63.2%)			160 (30.4%)			34 (6.5%)		
*p*-value	*p* < 0.05			*p* < 0.05			*p* < 0.05		

Abbreviations: CMR, cardiac magnetic resonance; PET, positron emission tomography; SPECT, single-photon emission computed tomography.

**Table 3 medicina-58-00368-t003:** Comparison of two groups (1 group—viable myocardium, reversibility score > 10; 2 group—non-viable myocardium, reversibility score ≤ 10).

	Group 1 N = 17	Group 2 N = 14	*p*-Value
Mean score CMR LGE	28.8 ± 9.6	28.2 ± 6.8	0.64
Reversibility Score (CMR 4 vs. PET 3/2/1/0)	14.4 ± 3.0	5.9 ± 2.8	<0.05
Age (years)	59.4 ± 9.2	62.4 ± 9.7	0.54
BMI (kg/m^2^)	28.9 ± 6.8	28.9 ± 2.6	0.72
6 MWT (m)	389.2 ± 94.5	301.4 ± 48.2	<0.05
LV EDD (TTE) (mm)	53.2 ± 7.9	63.4 ± 8.9	<0.05
MM (TTE) (g)	239.5 ± 85.9	276.3 ± 62.7	<0.05
LV EF (TTE) (%)	31.5 ±8.0	26.5 ± 7.8	0.24
RV (TTE) (mm)	33.4 ± 6.9	38.5 ± 5.0	<0.05
TAPSE (TTE) (mm)	18.7 ± 2.0	15.2 ± 2.0	<0.05
LVEDD (CMR) (mm)	61.7 ± 8.1	69.0 ± 6.1	<0.05
LVEDDi (CMR) (mm/m^2^)	29.8 ± 3.7	35.2 ± 3.1	<0.05
LVEDV (CMR) (ml)	282.9 ± 95.7	313.7 ± 114.4	0.43
LVEDVi (CMR) (ml/m^2^)	137.5 ± 48.5	156.7 ± 53.7	0.16
LV EF (CMR) (%)	34.1 ± 10.3	29.5 ± 9.3	0.27
LV GLS (CMR)	−14.3 ± 2.1	−11.4 ± 2.9	<0.05
LV GCS (CMR)	−17.2 ± 4.6	−12.7 ± 2.6	<0.05
Infarct size (g) (CMR)	40.4 ± 15.7	54.2 ± 25.0	0.62
Infarct size (%) (CMR)	24.5 ± 9.6	34.8 ± 11.1	<0.05

Continuous variables are expressed as mean (M) ± standard deviation (SD) for symmetric data. Categorical variables expressed as count and percentage of patients. Abbreviations: CMR LGE, cardiac magnetic resonance late gadolinium enhancement; PET, positron emission tomography; BMI, body mass index; 6 MWT, six-minute walk test; LV EDD, left ventricle end-diastolic diameter; TTE, transthoracic echocardiography; MM, myocardial mass; LV EF, LV ejection fraction; RV, right ventricle; TAPSE, tricuspid annular plane systolic excursion; LVEDDi, LVEDD index; LVEDV, LVED volume; LV global longitudinal strain; LV GCS, global circumferential strain.

**Table 4 medicina-58-00368-t004:** Correlation of different clinical and imaging parameters with the reversibility score (CMR 4 score vs. PET 3/2/1/0 score).

	Correlation Coefficient
6 MWT	rS = 0.48
LV EDD (TTE)	r = −0.70
MM (TTE)	r = −0.55
LV EF (TTE)	r = 0.46
RV (TTE)	r = −0.36
TAPSE (TTE)	r = 0.60
LVEDD (CMR)	r = −0.76
LVEDDi (CMR)	rS = −0.72
LVEDV (CMR)	r = −0.64
LVEDVi (CMR)	r = −0.60
LV EF (CMR)	rS = 0.39
LV GLS (CMR)	r = −0.64
LV GCS (CMR)	rS = −0.72
Infarct size (g) (CMR)	rS = −0.52
Infarct size (%) (CMR)	r = −0.61

r—Pearson correlation coefficient; rS—Spearman‘s rank correlation coefficient. All correlations in Table 4 are significant, *p* < 0.05. Abbreviations: CMR, cardiac magnetic resonance; PET, positron emission tomography; 6 MWT, six-minute walk test; LV EDD, left ventricle end-diastolic diameter; TTE, transthoracic echocardiography; MM, myocardial mass; LV EF, LV ejection fraction; RV, right ventricle; TAPSE, tricuspid annular plane systolic excursion; LVEDDi, LVEDD index; LVEDV, LVED volume; LV global longitudinal strain; LV GCS, global circumferential strain.

**Table 5 medicina-58-00368-t005:** Linear regression model when the independent variable is reversibility score (CMR 4 score vs. PET 3/2/1/0 score) and coefficient of determination (R^2^) is 0.72.

	B, 95% CI	*p*-Value
Constant	25.8 [8.2–43.4]	<0.05
LVEDD (CMR)	−0.4 [0.5–0.2]	<0.05
LV EF (CMR)	−0.2 [0.4–0.0]	<0.05
LV GCS (CMR)	−0.9 [1.5–0.4]	<0.05

Abbreviations: CMR, cardiac magnetic resonance; PET, positron emission tomography; CI, confidence interval; LV EDD, left ventricle end-diastolic diameter; LV EF, LV ejection fraction; LV GCS, global circumferential strain.

**Table 6 medicina-58-00368-t006:** ROC analysis.

	Group 1 N = 17 (100%)	Group 2 N = 14 (100%)	OR, 95% CI	AUC	Sensitivity	Specificity	*p*-Value
6 MWT (m) >350 ≤350	9 (52.9) 8 (47.1)	1 (7.1) 13 (92.9)	14.6 [1.5–138.2]	77.1	92.9	52.9	<0.05
LV EDD (TTE) (mm) <60 >60	13 (76.5) 4 (23.5)	5 (35.7) 9 (64.3)	5.9 [1.2–28.0]	88.7	64.3	76.5	<0.05
MM (TTE) (g) <240 >240	10 (58.8) 7 (41.2)	3 (21.4) 11 (78.6)	5.2 [1.1–26.0]	75.6	100	47.1	<0.05
RV (TTE) (mm) <34 >34	10 (58.5) 7 (41.2)	1 (7.1) 13 (92.9)	18.6 [2.0–176.5]	82.8	92.9	70.6	<0.05
TAPSE (TTE) (mm) >15 ≤15	16 (94.1) 1 (5.9)	5 (35.7) 9 (64.3)	28.8 [2.9–286.4]	84.5	64.3	94.1	<0.05
LVEDD (CMR) (mm) <64 >64	16 (94.1) 1 (5.9)	1 (7.1) 13 (92.9)	208.0 [11.8–3656.8]	97.1	92.9	94.1	<0.05
LVEDDi (CMR) (mm/m^2^) <32 >32	12 (70.6) 5 (29.4)	0 (0) 14 (100)	0.3 [0.1–0.6]	95.8	71.4	100	<0.05
LVEDV (CMR) (ml) <330 >330	14 (82.4) 3 (17.6)	6 (42.9) 8 (57.1)	0.2 [0.0–0.8]	71.2	57.1	88.2	<0.05
LVEDVi (CMR) (ml/m^2^) <140 >140	13 (76.5) 4 (23.5)	5 (35.7) 9 (64.3)	0.2 [0.0–0.8]	76.9	64.3	82.4	<0.05
LV EF (CMR) (%) >40 ≤40	7 (41.2) 10 (58.8)	1 (7.1) 13 (92.9)	9.1 [1.0–86.5]	64.5	92.9	41.2	<0.05
LV GLS (CMR) >−12.1 <−12.1	16 (94.1) 1 (5.9)	6 (42.9) 8 (57.1)	21.3 [2.2.–208.3]	81.9	57.1	100	<0.05
LV GCS (CMR) >−15.8 <−15.8	11 (64.7) 6 (35.3)	2 (14.3) 12 (85.7)	11.0 [1.8–66.4]	83.6	85.7	70.6	<0.05
Infarct size (g) (CMR) <48 >48	15 (88.2) 2 (11.8)	6 (42.9) 8 (57.1)	0.1 [0.0–0.6]	80.9	57.1	100	<0.05
Infarct size (%) (CMR) <26 >26	12 (70.6) 5 (29.4)	4 (28.6) 10 (71.4)	0.2 [0.0–0.8]	78.2	71.4	88.2	<0.05

Abbreviations: ROC, receiver operating characteristic; OR, odds ratio; CI, confidence interval; AUC, the area under the curve; CMR, cardiac magnetic resonance; PET, positron emission tomography; 6 MWT, six-minute walk test; LV EDD, left ventricle end-diastolic diameter; TTE, transthoracic echocardiography; MM, myocardial mass; LV EF, LV ejection fraction; RV, right ventricle; TAPSE, tricuspid annular plane systolic excursion; LVEDDi, LVEDD index; LVEDV, LVED volume; LV global longitudinal strain; LV GCS, global circumferential strain.

**Table 7 medicina-58-00368-t007:** Binary logistic regression model for the prediction that the reversibility score is < 10, Nagelkerke R^2^ 0.57; overall percentage 80.6.

	B, 95% CI	*p-*Value
LVEDV > 330 mL (CMR)	2.0 [0.9–68.2]	0.07
Infarct size > 26% (CMR)	2.2 [1.1–69.4]	<0.05
LV GCS < −15.8 (CMR)	2.5 [1.4–102.3]	<0.05

Abbreviations: CMR, cardiac magnetic resonance; LVEDV, left ventricle end-diastolic volume; LV GCS, global circumferential strain.

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
