# Peer review of "Comparative Analysis of Myocardial Viability Multimodality Imaging in Patients with Previous Myocardial Infarction and Symptomatic Heart Failure"

_medicina, 2022, doi:10.3390/medicina58030368_

Round 1
Reviewer 1 Report
Interesting study, however very low study sample limit generalizability of the results
- what new add this study to cuurent knowledge? Any new impact on clinical practise?
- Very low sample size is serious drwawback
- Results section "patients were suffering from dispone" do authors mean dyspnoe?
- How many physicians evaluated each imaging modality? There is risk for intraobserver varaibility and additional bias
Author Response
Point 1: what new add this study to cuurent knowledge? Any new impact on clinical practise?
Response 1: The results of our study suggest the clinical approach for myocardial viability evaluation, diminishing radiation exposure to the patient whenever it is possible firstly performing TTE and CMR. If the CMR results demonstrate a sufficient quantity of viable myocardium revascularisation should be performed. However, according to the results of our study, LGE CMR reveals a significantly higher number of scars, therefore FDG PET should be performed in cases when viability is doubtful in CMR, as FDG PET appears to be more optimistic in functional recovery prediction. This algorithm of myocardial viability evaluation could be applied in clinical practice, leading not only to the precise evaluation of myocardial viability guiding to the better clinical decision (medical treatment, revascularisation or heart transplantation) but also avoiding unnecessary radiation exposure to the patient.
Point 2: Very low sample size is serious drwawback
Response 2: The main limitation of our study was the small study sample. However, according to the study protocol, we have applied different imaging modalities for patients with ischaemic symptomatic HF and expanded coronary artery atherosclerosis when the need for revascularisation was doubtful due to suspected myocardial scar tissue. In this patient group, we were able to perform a head-to-head comparison of three imaging modalities (TTE, nuclear cardiology imaging and CMR) to evaluate viable myocardium. This approach was associated with time consumption, cost-effectiveness, radiation exposure and limited study population.
Point 3: Results section "patients were suffering from dispone" do authors mean dyspnoe?
Response 3: dispone->dyspnea
Point 4: How many physicians evaluated each imaging modality? There is risk for intraobserver varaibility and additional bias
Response 4: The CMR and nuclear cardiology imaging analysis were performed by one experienced physician dedicated to the exact imaging modality in our institution.
All the corrections are made to the manuscript (please see the attachment).
I sincerely appreciate your comments and suggestions.
Reviewer 2 Report
In the paper “ Comparative analysis of myocardial viability multimodality imaging in patients with previous myocardial infarction and symptomatic heart failure” authors compare three different methodic, SPECT MPI, PET and CMR, for the evaluation of left ventricle viability in patients with previous MI. The head-to-head test between different methodologies it is interesting and a very important adjunctive value of the paper. CMR resulted to be more sensitive, as expected, to identify scar where PET the more optimistic in term of functional recovery prediction.
The paper is well and clearly written; the discussion is logic and consequent to results found.
Nothing to change.
Thank you for the opportunity to review this interesting paper.
Author Response
I sincerely appreciate your comments.
Round 2
Reviewer 1 Report
no futher comments